# Augmented Conditioning is Enough for Effective Training Image Generation

## Abstract

Image generation abilities of text-to-image diffusion models have significantly advanced, yielding highly photo-realistic images from descriptive text and increasing the viability of leveraging synthetic images to train computer vision models. To serve as effective training data, generated images must be highly realistic while also sufficiently diverse within the support of the target data distribution. Yet, state-of-the-art conditional image generation models have been primarily optimized for creative applications, prioritizing image realism and prompt adherence over conditional diversity. In this paper, we investigate how to improve the diversity of generated images with the goal of increasing their effectiveness to train downstream image classification models, without fine-tuning the image generation model. We find that conditioning the generation process on an augmented real image and text prompt produces generations that serve as effective synthetic datasets for downstream training. Conditioning on real training images contextualizes the generation process to produce images that are in-domain with the real image distribution, while data augmentations introduce visual diversity that improves the performance of the downstream classifier. We validate augmentation-conditioning on a total of five established long-tail and few-shot image classification benchmarks and show that leveraging augmentations to condition the generation process results in consistent improvements over the state-of-the-art on the long-tailed benchmark and remarkable gains in extreme few-shot regimes of the remaining four benchmarks. These results constitute an important step towards effectively leveraging synthetic data for downstream training.

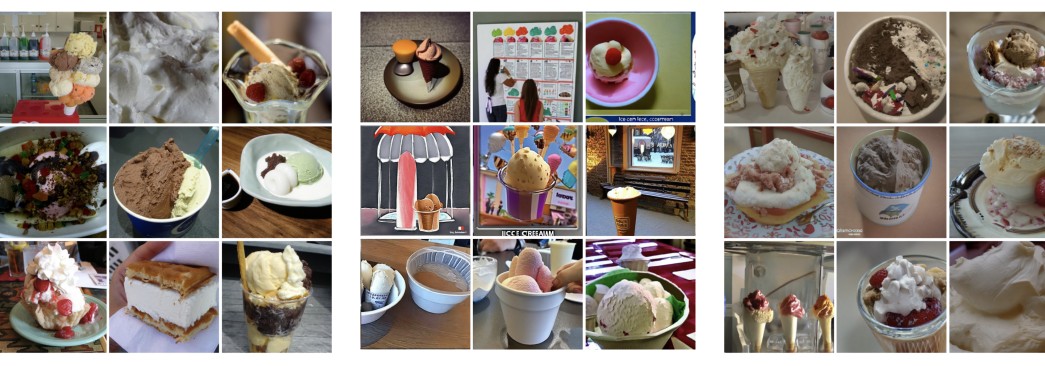

|   (a) ImageNet-LT   |   (b) Latent Diffusion   |   (c) Embed-CutMix-Dropout (Ours)   |

Figure 1: Example images from (a) real training data, (b) a pretrained diffusion model using the class label as conditioning, (c) the best performing augmentation-conditioned method. Augmentation conditioning generates visually diverse, realistic images that enhance downstream classification accuracy when used as training data.

## 1 Introduction

Advances in modern deep learning greatly rely on massive datasets. With the advent of large-scale pretraining and foundation models, massive amounts of diverse data are an integral part of AI.

---

[*]Authors A/B acted in an advisory role. None of the experiments were run on Institution A's infrastructure. The research was conducted by Author C prior to joining institution A.

State-of-the-art datasets have only increased in size with time; from ImageNet-1k Deng et al. (2009) consisting of 1.3 million images from 1000 classes, to the current LAION dataset Schuhmann et al. (2022) that consists of 5 billion image-caption pairs from the Internet. Particularly in computer vision, high-quality images that are diverse and in-domain are crucial to classification performance. However, collecting real images is often expensive or difficult; especially in specialized tasks where examples of classes are rare or hard to photograph. This leads to long-tail, imbalanced classification settings where most classes have very few training examples (Liu et al., 2019; Ren et al., 2020; Kang et al., 2020). Additionally it is well-known that visual diversity, traditionally introduced through data augmentation on existing training images, improves classifier performance and generalization (Krizhevsky et al., 2017; Yun et al., 2019; Zhang et al., 2018; Cubuk et al., 2019).

Recently, diffusion text-to-image models have achieved unprecedented standards for synthetic image quality, capable of generating photo-realistic images for an impressive variety of text prompts (Podell et al., 2023; Ramesh et al., 2022; Saharia et al., 2022). A natural application for these models is synthetic training image generation, as the visual characteristics of generated images are controllable via various diffusion mechanics such as the conditioning information, guidance scale, and latent noise variables. However, diffusion models are primarily used to generate imaginative images from creative prompts rather than realistic depictions of real-world objects. Text-to-image models are often optimized for creativity purposes with human preference as a metric, prioritizing image quality and prompt adherence over generation diversity. This leads to synthetic images being less effective than real images when used as training data, as synthetic images often depict spurious qualities of image classes and have style bias from their training dataset (He et al., 2023; Sariyildiz et al., 2023). Furthermore, training images must be visually diverse to increase classification performance and properly represent variations of visual concepts, but pretrained diffusion models often lack the ability to generate images that reflect the representation diversity found in real-world domains (Dunlap et al., 2023; Trabucco et al., 2023; Luccioni et al., 2023; Wan et al., 2024; Hall et al., 2023).

Existing methods for training image generation remedy these issues by fine-tuning the diffusion model on task-specific data Azizi et al. (2023), using large language models to prompt for diversity in image generations Dunlap et al. (2023), or using specialized fine-tuning of the diffusion model to learn concepts from real training images (Shin et al., 2023; Trabucco et al., 2023). However, fine-tuning of diffusion models is computationally expensive, especially when the classification task has many visual concepts the diffusion model must learn.

In this paper, we analyze the use of classical vision data augmentation methods as conditioning information for image generation and find certain data augmentations yield visually diverse training images that enhance downstream classification. We use augmentation-conditioning and a frozen, pretrained diffusion model to generate effective training images in a much more computationally efficient manner than previous work that requires diffusion model training *e.g.*, (Azizi et al., 2023; Trabucco et al., 2023; Shin et al., 2023). In particular, augmentation-conditioning leverages vision data augmentations of real images alongside a text prompt as conditioning information in the image generation process. Conditioning on real training images provides in-domain context to the generation process whereas the proposed use of data augmentations encourage visual diversity, altogether increasing the performance of downstream classification while requiring the same computational cost as off-the-shelf image generation with a pretrained diffusion model. We evaluate various augmentation methods on five ubiquitous long-tail and few-shot classification tasks, in both training from scratch and fine-tuning settings, showing that our synthetic datasets improve classification performance over existing work.

We find that that using augmentation-conditioned synthetic datasets results in outperforming prior work on ImageNet Long-Tailed, while training on 135k less synthetic images. Augmentation conditioning also enables surpassing state-of-the-art classification accuracy on four standard few-shot benchmarks and exhibits remarkable gains in extreme few-shot regimes, even when compared to methods that require diffusion model training or finetuning. These results highlight the potential of augmentation-conditioned techniques to generate training data, without requiring any generative model finetuning, and constitute an important step towards effectively leveraging synthetic data for downstream model training.

## 2 RELATED WORK

**Synthetic Training Data from Generative Models.** Early work used class-conditioned Generative Adversarial Networks (GANs) to generate synthetic training images (Besnier et al., 2019; Li

et al., 2022; Ravuri & Vinyals, 2019). More recently as diffusion has become dominant for image generation, most works utilize text-to-image diffusion models for synthetic training data. Previous works using diffusion models has found that only using text class labels for image generation results in synthetic training datasets that cannot match the performance of real image datasets, mainly due to domain gap between real and synthetic images (He et al., 2023; Sariyildiz et al., 2023). The domain gap issue is somewhat remedied by fine-tuning the diffusion model on real images (Azizi et al., 2023). However, fine-tuning diffusion models is computationally expensive or infeasible in classification settings where real images of class concepts are rare.

**Diffusion-Based Image Augmentations.** Promising classification results have been shown in existing work that uses diffusion models to edit or augment real images rather than fully generate synthetic images. These methods use diffusion models to introduce visual diversity to real images then perform few-shot fine-tuning of pretrained classifiers on generated images. Existing work has used a large language model to guide diffusion model image editing Dunlap et al. (2023) or used textual inversion Gal et al. (2022) to fine-tune the diffusion model and learn realistic representations of classes for each image generation (Trabucco et al., 2023). Inspired by these diffusion augmentation methods, we experiment with conditioning diffusion on augmented real images, rather than using diffusion to augment images. This avoids the expensive fine-tuning of the diffusion model or using models other than the image generator, but still introduces visual diversity by leveraging classical vision augmentations.

**Synthetic Images for Long-Tail Classification.** Long-tail classification is the setting where most training classes have few examples, and additionally the examples per class are imbalanced but the test set is balanced. This classification setting occurs in the real world when class concepts are rare or difficult to photograph (Horn et al., 2018; Liu et al., 2019). Many methods not involving synthetic training data have approached this problem with various training loss and representation learning approaches (Kang et al., 2020; Ren et al., 2020; Liu et al., 2019). We apply augmentation-conditioned generations to long-tail classification, to explore their efficacy as training data when training classifiers from scratch.

To our knowledge, only two other works have applied diffusion-based image generation to long-tail classification benchmarks. Shin et al. (2023) uses textual inversion Gal et al. (2022), a training technique that teaches the diffusion model about a particular visual concept from the real training images, to balance the amount of training images per-class. Hemmat et al. (2023) also balances the number of training images for each class with synthetic images; their generation method uses classification performance from a separate, pretrained classifier in the diffusion guidance term as well as conditions on the text class label and a real training image. Du et al. (2023) uses traditional vision augmentations (without a diffusion model) to create training data, however our method outperforms it.

## 2.1 Data Augmentation in Computer Vision

Image augmentation has long been a core component of training deep vision models, known to reduce overfitting and encourage generalization (Krizhevsky et al., 2017; Cubuk et al., 2019; Zhang et al., 2018; Yun et al., 2019). A variety of existing augmentations that leverage color and geometric transformations on images are known to increase classification robustness on vision benchmark datasets and are considered a standard part of training. Various image translations and reflections as well as altering RGB intensities are effective for ImageNet (Krizhevsky et al., 2017). CutMix, i.e. randomly cutting and pasting pixels between training images while proportionally mixing image labels, is an effective localized augmentation method (Yun et al., 2019). Mixup, i.e. convex combinations of images and their labels, is a form of data interpolation that increases robustness to adversarial examples and training stability of generative adversarial networks (Zhang et al., 2018). More recently, the learned augmentation method RandAugment, which composes various geometric and color transformations, has become widely used in vision (Cubuk et al., 2019). We leverage CutMix and MixUp in the conditioning information of diffusion, which effectively introduces diversity to our generations. One of our few-shot baselines is a direct comparison to data generated via RandAugment.

## 3 Augmentation-Conditioned Generations

Generations must be in-domain and realistic to facilitate effective classifier learning, to enforce this we condition the diffusion process on real training images. Visually diverse training data adds robustness to classification, and we leverage data augmentations in the conditioning information of the diffusion process to make our generations more diverse. Given labeled training images, we apply vision augmentations and use the augmented images as conditioning information for the diffusion

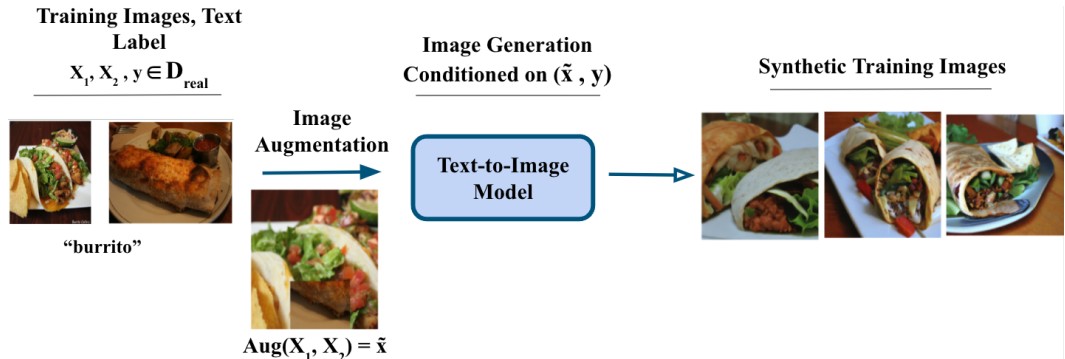

Figure 2: Our augmentation-conditioned generation conditions the reverse diffusion process on the class label and an augmented real image, introducing visual diversity that improves the performance of the downstream classifier.

process, resulting in synthetic images that are visually diverse while still in-domain with real images. We apply and ablate over various augmentations to explore which are most effective in various training settings. Figure 2 shows an overview of the augmentation-conditioned generation process.

### 3.1 ENSURING GENERATIONS ARE IN-DOMAIN WITH CONDITIONING

Generating images using only the text class labels and no fine-tuning of the diffusion model is known to result in images with semantic issues that lessen their effectiveness as training data (Sariyildiz et al., 2023; Hemmat et al., 2023; He et al., 2023). Additionally, using learned or manual prompt engineering based on class names is unable to yield classification performance on par with real images (Sariyildiz et al., 2023; He et al., 2023). We identify specific failure cases where using only class names for generations results in synthetic images out of the domain of real classification data: **1) Semantic Errors**, where synonyms and homonyms in class labels lead to images of objects that do not exist in the real training set; **2) Visual Domain Shift**, where style bias from the diffusion model's training data results in generations of a distinctly different visual style. Training classifiers on data exhibiting these failure cases are greatly detrimental to classification performance.

To remedy these issues, we follow Hemmat et al. (2023) and condition image generation on both the text class label and a real training image of the corresponding class. This approach is simpler and yields better classification results than existing approaches that utilize prompt engineering or generating prompts with LLMs (Sariyildiz et al., 2023; Dunlap et al., 2023). Additionally, pre-trained image-conditioned or image variation diffusion models are commonly available (HuggingFace, 2023; von Platen et al., 2022), making this approach is easily accessible. As seen in Figure 3, simply conditioning on a randomly selected training image from the text class label alleviates failure cases.

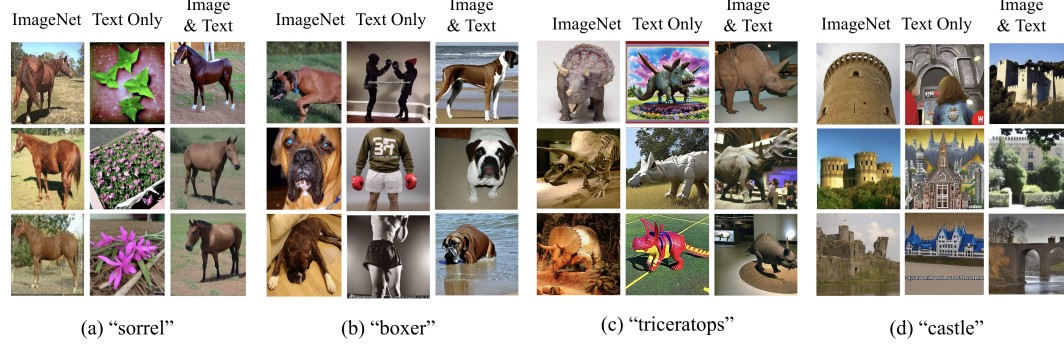

(a) "sorrel"    (b) "boxer"    (c) "triceratops"    (d) "castle"

Figure 3: Failed generations: (a), (b) **Semantic Errors**, where generations using only the class label result in images depicting a totally different object; (c), (d) **Visual Domain Shift**, where generations using only the class label produce the correct visual concept but in a distinctly different visual style. Both these failure cases reduce efficacy of synthetic training images and are remedied by generating images conditioned on the class label and real training images.

**Conditioning on**

(a) "hamster" and real image:

(b) "hamster" and real images:

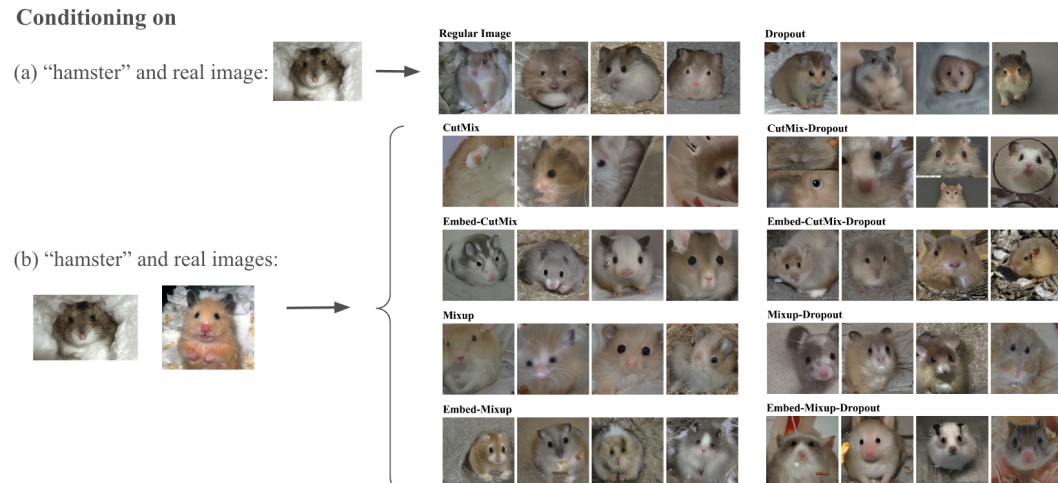

Figure 4: Sample generated images using all of the augmentation conditioning methods. (a) shows baseline generations conditioned on an original training image and generations conditioned on Dropout applied to the training image (b) shows generations conditioned on the combination of 2 training images produced with the specified augmentation method. *Augmentation-conditioned generations show more visual diversity in the coloration, pose, and angle of the hamster* compared to the Regular Image generation. Generations from Embed-CutMix-Dropout, which yields the highest accuracy on ImageNet-LT, have distinct background diversity with hamsters depicted in various realistic terrains.

However, introducing image conditioning reduces visual diversity of generations, which we address in the next section.

## 3.2 ADDING VISUAL DIVERSITY TO IN-DOMAIN GENERATIONS

Inspired by traditional vision, we use image augmentation methods to introduce diversity into our generations. Augmentations are applied to real images, in both pixel and embedding space, then diffusion is conditioned on the augmented data and the text class label. The diffusion model we use, a latent diffusion model (LDM) conditioned on image and text features referred to as LDM-v2.1-unCLIP (HuggingFace, 2023), encodes the conditioning image into the CLIP (Radford et al., 2021) embedding space before conditioning, enabling us to perform augmentations in CLIP embedding and pixel space. We leverage the well-known CutMix (Yun et al., 2019) and Mixup (Zhang et al., 2018) augmentations on 2 randomly selected training images of the same class $x_1, x_2$:

$$\text{CutMix:} \quad \tilde{x} = \mathbf{M} \odot x_1 + (\mathbf{1} - \mathbf{M}) \odot x_2$$
$$\text{Mixup:} \quad \tilde{x} = \lambda x_1 + (1 - \lambda) x_2$$

For CutMix, $\mathbf{M}$ is a binary mask sampled based on $\lambda$ indicating where to replace an image region of $x_1$ with a patch from $x_2$, $\mathbf{1}$ is a binary mask of all ones, and $\odot$ is element-wise multiplication. For Mixup and CutMix, $\lambda$ is sampled from a Beta distribution with $\alpha = 1.0$, the default setting in `torchvision`. If the augmentation is done in pixel space then $x_1, x_2$ are images and the resulting $\tilde{x}$ is later encoded into a CLIP image embedding; if the augmentation is done in embedding space then $x_1, x_2$ are CLIP image embeddings of the corresponding images and $\tilde{x}$ is a combined embedding.

We also use Dropout (Srivastava et al., 2014) with $p = 0.4$, on the CLIP image embedding of a randomly selected training image, as a stochastic augmentation method that removes random parts of the image conditioning information. This is equivalent to using a Dropout layer on the last layer of the CLIP image encoder. As seen in Figure 7, we observe that the Dropout probability acts as an image generation hyperparameter controlling the conditioning strength of the text and image information, with $p = 0.0$ resulting in homogeneous images all similar to the conditioning image and $p = 1.0$ resulting in images exhibiting failure cases discussed in Section 3.1. Thus, an intermediate Dropout ratio results in the most visually diverse generations, given the same conditioning text and image.

A total of 9 augmentation-conditioned methods result from combinations of the aforementioned augmentation methods: Dropout, CutMix, CutMix-Dropout, Embedding-CutMix, Embedding-CutMix-Dropout, Mixup, Mixup-Dropout, Embedding-Mixup, and Embedding-Mixup-Dropout. For the

combination methods, we perform CutMix or Mixup in the specified pixel or embedding space then apply Dropout to the augmented embedding. Let $\tilde{x}$ be the image embedding produced by an augmentation method; to condition the image generation process on the augmentation, the diffusion denoising UNet (Ronneberger et al., 2015) concatenates $\tilde{x}$ onto its time step embedding. Sample generations for all conditioning methods are shown in Figure 4.

# 4 EXPERIMENTS

We generate synthetic training datasets with each augmentation-conditioning method in Section 3.2 and evaluate the efficacy of each image augmentation method by training downstream classifiers on images generated using the augmentation as conditioning information. We show the efficacy of augmentation-conditioned generations in two settings: (1) training from scratch in a large scale, long-tail setting with class-imbalanced classification and (2) fine-tuning a pre-trained classifier on various few-shot classification tasks.

## 4.1 LARGE-SCALE IMBALANCED CLASSIFICATION

**Class-Imbalanced, Long-tail Dataset.** Augmentation-conditioned generations are naturally applicable to long-tailed data settings, where examples per class are imbalanced and most classes have scarce examples. We use augmented existing real examples as conditioning information and generate synthetic images to balance the number of examples across classes, then train a downstream classifier on the combined set of synthetic and real images and evaluate on a balanced test set of real images.

Our experiments use the largest and most ubiquitous long-tail benchmark dataset, ImageNet-LT (Liu et al., 2019), a subset of ImageNet-1K (Deng et al., 2009) downsampled so that most classes have around 20 training images. ImageNet-LT has a total of 115.8k real images across 1K classes, with a minimum of 5 and maximum of 1,280 images per class. Classes are categorized based on their number of training examples: many-shot for 100 or more, medium-shot for 20 to 100, and few-shot for 20 or less. We generate enough synthetic images so that each class has 1,280 training images, resulting in a total of approximately 1.16 million synthetic images.

**Experimental Setup.** For image generation, we use the pre-trained LDM-v2.1-unCLIP model (HuggingFace, 2023). This model is based on LDM v2.1 (Rombach et al., 2022) and is capable of generating images conditioned on text and image. We use this diffusion model off-the-shelf with no changes to its weights. In line with previous work on ImageNet-LT, we train a ResNext50 (Xie et al., 2016) classifier from scratch for 150 epochs using the SGD optimizer with cosine annealing (Loshchilov & Hutter, 2016) and the Balanced Softmax loss (Ren et al., 2020). We measure efficacy of augmentation-conditioned synthetic training datasets by evaluating top-1 accuracy on the balanced test set of real images. During training each minibatch contains 50% real and 50% synthetic images, as this balancing of real and synthetic images is known to improve training stability (Hemmat et al., 2023; Trabucco et al., 2023; He et al., 2023). For full details on image generation and training hyperparameters see Appendix B.

### 4.1.1 CONDITIONING METHOD PERFORMANCE

To initially compare the performance of our nine augmentation-conditioned generation methods under compute constraints, we ran smaller scale evaluations on 90 randomly selected classes of ImageNet-LT with a ResNet18 classifier. This class subset includes 30 of each of the few, median, and many class categories. Overall accuracies as well as class category accuracies on the corresponding 90-class-subset evaluation set are reported in Table 1.

The conditioning method using CutMix and Dropout in the CLIP embedding space performs best, followed closely by embedding-space Mixup and Dropout, and solely Dropout. Conditioning using embedding-space CutMix and Dropout enables about +4% overall accuracy over conditioning on an un-augmented training image (Random Image in Table 1) and a remarkable +8% accuracy on the hardest category of few-shot classes. Dropout done in addition to any of the image augmentation methods, regardless of in pixel of embedding space, increases accuracy; indicating that Dropout as a data augmentation yields effective conditioning information for synthetic training image generation.

We calculate Fréchet Inception Distance (FID) Score (Chong & Forsyth, 2019), a measure of both image quality and diversity, between the evaluation set of real images and the synthetic training dataset for each of the augmentation-conditioned generation methods. The best-performing augmentation-

Table 1: Top-1 classification accuracy and FID Score between synthetic datasets and evaluation set for ImageNet-LT 90-class-subset. Conditioning Methods are discussed in Section 3.2; Random Image is a baseline generation conditioned on the class label and a randomly selected training image of that class. The best accuracy per-category is bolded.

| Conditioning Method | Overall | Many | Median | Few | FID Score |
|---|---|---|---|---|---|
| Random Image (Baseline) | 63.0 | 72.4 | 61.4 | 55.3 | 20.181 |
| Dropout | 66.2 | 70.9 | 64.7 | 63.0 | 21.843 |
| Mixup | 63.6 | 69.5 | 63.3 | 58.0 | 24.115 |
| Mixup-Dropout | 65.6 | 69.2 | 65.2 | 62.4 | 22.306 |
| Embed-Mixup | 63.5 | 71.3 | 62.4 | 56.8 | 22.930 |
| Embed-Mixup-Dropout | 66.2 | 72.2 | 63.7 | 62.7 | 24.558 |
| CutMix | 63.8 | 69.5 | 63.0 | 59.0 | 26.623 |
| CutMix-Dropout | 65.2 | 69.2 | 63.1 | 63.2 | 24.453 |
| Embed-CutMix | 62.6 | **73.1** | 61.9 | 53.0 | 20.285 |
| **Embed-CutMix-Dropout** | **66.9** | 72.0 | **65.2** | **63.5** | 20.433 |

conditioning method has one of the lowest FID scores, supporting our claim that augmentation-conditioned generations increase *in-distribution diversity* and lead to better classification performance.

### 4.1.2 CLASSIFIER FREE GUIDANCE SCALE

The classifier free guidance (CFG) scale parameter of diffusion models controls the trade-off between prompt adherence and diversity of generations (Ho & Salimans, 2022). Previous work on synthetic training image generation found that the CFG scale greatly affects downstream classification accuracy, with lower values leading to better performance empirically (Fan et al., 2023; Tian et al., 2023; Sariyildiz et al., 2023). To explore CFG scale's effect on augmentation-conditioned generations, we run the best-performing conditioned generation method Embed-CutMix-Dropout with CFG scales: [2.0, 4.0, 7.0, 10.0] and report maximum validation accuracy over all epochs on the 90-class-subset in Table 2.

Table 2: Classifier Free Guidance (CFG) scale's effect on top-1 classification validation accuracy on ImageNet-LT 90-class-subset. The lowest CFG scale of 2.0 results in highest overall accuracy.

| CFG Scale | Overall | Many | Median | Few |
|---|---|---|---|---|
| 2.0 | **73.3** | **75.5** | 72.0 | **72.3** |
| 4.0 | 72.9 | 75.3 | **72.2** | 71.2 |
| 7.0 | 70.5 | 74.5 | 68.5 | 68.5 |
| 10.0 | 66.9 | 72.0 | 65.2 | 63.5 |

The lowest CFG scale of 2.0 achieves the highest accuracy overall, with a notable almost +10% accuracy on the most difficult few-shot classes when compared to the `Hugging-Face` default CFG scale of 10.0. This result aligns with previous work which finds that a low CFG scale leads to the best downstream accuracy for ImageNet-scale synthetic training data, as it increases diversity across the numerous generations that use the same class text labels (Fan et al., 2023).

### 4.1.3 IMAGENET-LT BASELINES

We run the best four conditioning methods from the 90-class-subset results (Section 4.1.1) on full-scale ImageNet-LT, with results compared to existing baselines in Table 3. The augmentation-conditioning method using embedding-space CutMix and Dropout outperforms SOTA ImageNet-LT baselines that use no diffusion-generated images, though (Du et al., 2023) uses traditional vision augmentations to generate training data. It also outperform prior works that generate and train on similar quantities of synthetic data, improving accuracy over (Hemmat et al., 2023) with over 135k less synthetic images. These accuracy gains show that CutMix and Dropout augmentations in the CLIP embedding space provides valuable conditioning information that results in effective synthetic training data.

Note that Hemmat et al. (2023) proposes additional methods that use performance signals of a separate, pre-trained classifier in the diffusion process, which can improve upon our results but also incurs additional computation cost. Fill-Up (Shin et al., 2023) trains the classifier from scratch on over 2x the amount of synthetic training images we use and additionally fine tunes the classifier on real images after pre-training, so the comparison is unfair. Even with $2\times$ the synthetic data amount and fine-tuning, Fill-Up only achieves +4% accuracy over the best augmentation-conditioned method. Previous work (Fan et al., 2023) has found that classification accuracy increases as the amount of

Table 3: Top-1 classification accuracy on ImageNet-LT using ResNext50. The best augmentation-conditioning method outperforms SOTA accuracy of methods that use no synthetic data. We outperform methods utilizing similar amounts of synthetic data, while Fill-Up (which uses more than 2x the amount of synthetic training images and fine-tunes the model on real images after pre-training) only outperforms us by less than 4%.

| Method | Synthetic Data Count | ImageNet-LT | | | |
|---|---|---|---|---|---|
| | | Overall | Many | Medium | Few |
| Decouple-LWS (Kang et al., 2020) | 0 | 47.7 | 57.1 | 45.2 | 29.3 |
| Balanced Softmax (Ren et al., 2020) | 0 | 51.0 | 60.9 | 48.8 | 32.1 |
| Mix-Up GLMC  (Du et al., 2023) | 0 | **57.21** | **64.76** | **55.67** | **42.19** |
| Fill-Up (Shin et al., 2023) | *2.6M* | 63.7 | 69.0 | 62.3 | 54.6 |
| LDM (txt) (Hemmat et al., 2023) | 1.3M | 57.9 | 64.8 | 54.6 | 50.3 |
| LDM (txt and img) (Hemmat et al., 2023) | 1.3M | 58.9 | 56.8 | 64.5 | 51.1 |
| Dropout (Ours) | 1.16M | 57.3 | 65.8 | 54.3 | 44.0 |
| Mixup-Dropout (Ours) | 1.16M | 57.4 | 65.8 | 53.9 | 46.3 |
| Embed-Mixup-Dropout (Ours) | 1.16M | 56.0 | 65.3 | 52.4 | 42.2 |
| Embed-CutMix-Dropout (Ours) | 1.16M | **59.6** | **66.3** | **56.6** | **51.1** |

synthetic images scales, so we can expect the accuracy gap to be closed if we generated and trained on more synthetic images; but due to compute constraints, we were unable to run experiments with more generated images.

## 4.2 FEW-SHOT CLASSIFICATION

**Few-Shot Vision Datasets.**   In line with previous diffusion-augmentation work, we benchmark augmentation-conditioned generations on four computer vision datasets: Caltech101 (Fei-Fei et al., 2004), Flowers102 (Nilsback & Zisserman, 2008), COCO (Lin et al., 2014) (2017 version), and Pascal VOC (Everingham et al., 2010) (2012 version). Pascal VOC and COCO are originally object detection datasets, but we adapt them into classification datasets by using the class label of the object with the largest pixel mask as the image label, as is done in previous work we use as baseline comparisons (Trabucco et al., 2023). By this labelling method, COCO has 80 classes and Pascal VOC has 20 classes. Caltech101 and Flowers102 each have 102 classes. Caltech101, Pascal VOC, and COCO have common classes (e.g. "car", "cat") and Flowers102 has only niche, fine-grained classes which are flower species (e.g. "alpine sea holly").

**Experimental Setup.**   We use the same diffusion model from the previous section's class-imbalanced experiments, LDM-v2.1-unCLIP (HuggingFace, 2023). A ResNet50 (He et al., 2015) pre-trained in ImageNet is fine-tuned on a mixture of real and synthetic images, where each image in a minibatch has a 50% probability of being a real training image and 50% probability of being a synthetic image, as done in (Trabucco et al., 2023). We fine-tune the last layer of the ResNet50 for 50 epochs using the Adam optimizer and a learning rate of 0.0001. To match the accuracies reported in (Trabucco et al., 2023), we report the highest validation accuracy across epochs. Additionally, we run fine-tuning with 1, 2, 4, 8, and 16 examples per class in the training set, and report mean validation accuracy over 4 independent trials. Points in our plots represent accuracy means and shading represents variance; though most variance values are in the $10^{-6}$ range and therefore not visible. [1]

The baselines we compare to are taken directly from Trabucco et al. (2023) and include three different data augmentation methods. RandAugment (Cubuk et al., 2019) is a widely used augmentation method involving color and geometric transformations that uses no generated images. Real Guidance (He et al., 2023) generates synthetic images using SDEdit (Meng et al., 2021), i.e. noising a real image, then denoising the noised image with a stochastic differential equation prior. DA-Fusion (Trabucco et al., 2023) generates synthetic images by training the diffusion model to learn the class's visual concept from real training images via textual inversion (Gal et al., 2022) and additionally uses SDEdit at image generation time. Note that our augmentation-conditioning methods require significantly less computation and memory than DA-Fusion, as they require no changes to the diffusion model but DA-Fusion requires training the diffusion model for each generated image.

---

[1]We cannot plot variance for results from existing work in Figure 6 due to compute constraints and the unavailability of raw results from the authors.

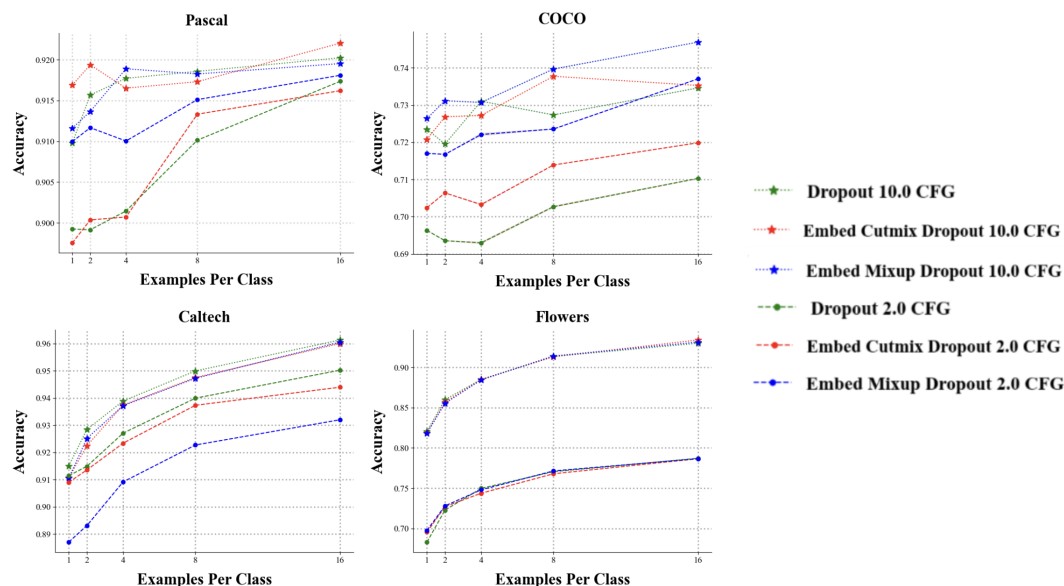

Figure 5: Classifier free guidance scale's effect on few-shot classification performance. Across all datasets, fine-tuning on images generated with 10.0 CFG scale yields better performance.

### 4.2.1 CLASSIFIER FREE GUIDANCE SCALE

As discussed and seen in the results of Section 4.1.2, the Classifier Free Guidance (CFG) scale parameter of image generation has notable effect on the synthetic images and downstream accuracy. We explore if CFG scale still has an effect when fine-tuning on a relatively small amount of synthetic data by running the same fine-tuning experiments on images generated with a CFG scale of 2.0 (the optimal CFG scale for ImageNet-LT) and 10.0 (the default CFG scale for our diffusion model), with results in Figure 5. We use the conditioning methods with the top 3 accuracies from the experiments in Section 4.1.1, and more detailed individual plots are in Appendix C.

Interestingly, for all datasets the optimal CFG scale for fine-tuning is not the optimal CFG scale for large-scale training from scratch. The same conditioning methods used with the 10.0 CFG scale yield higher few-shot accuracies than when used with the 2.0 CFG scale across all four datasets. We believe this is because the few-shot setting uses very few synthetic images compared to large-scale training, so strong prompt adherence and high image quality is more important to the classifier's learning than visual diversity.

### 4.2.2 FEW-SHOT BASELINES

Figure 6 shows that augmentation-conditioned generation methods improve accuracy across all datasets. We applied the the conditioning methods with the top 3 accuracies from Section 4.1.1's experiments, and plot the augmentation-conditioned method that yielded the highest few-shot accuracy per-dataset (all augmentation-conditioned method performance can be seen in Figure 5).

Augmentation-conditioned generations match or yield up to +25% accuracy over the best-performing existing method DA-Fusion (Trabucco et al., 2023), which requires training of the diffusion model whereas augmentation-conditioning requires no training. For the Pascal VOC and Flowers102 datasets, augmentation-conditioned augmentations outperforms all existing methods for all examples per class values, with approximately 10% higher accuracy for Pascal VOC and 3% for Flowers102. These performance gains indicate that augmentation-conditioning is effective at producing synthetic training images that are useful for fine-grained (e.g. flower species for Flowers102) and common object (e.g. general animal and vehicle types in Pascal VOC) classification, without requiring the diffusion model to learn visual concepts from real data.

## 5 DISCUSSION

**Conclusion.** We analyzed the efficacy of various vision data augmentation methods for synthetic training data generation via thorough experimentation, finding augmentation-conditioned generation capable of producing effective synthetic training datasets. Training on augmentation-conditioned

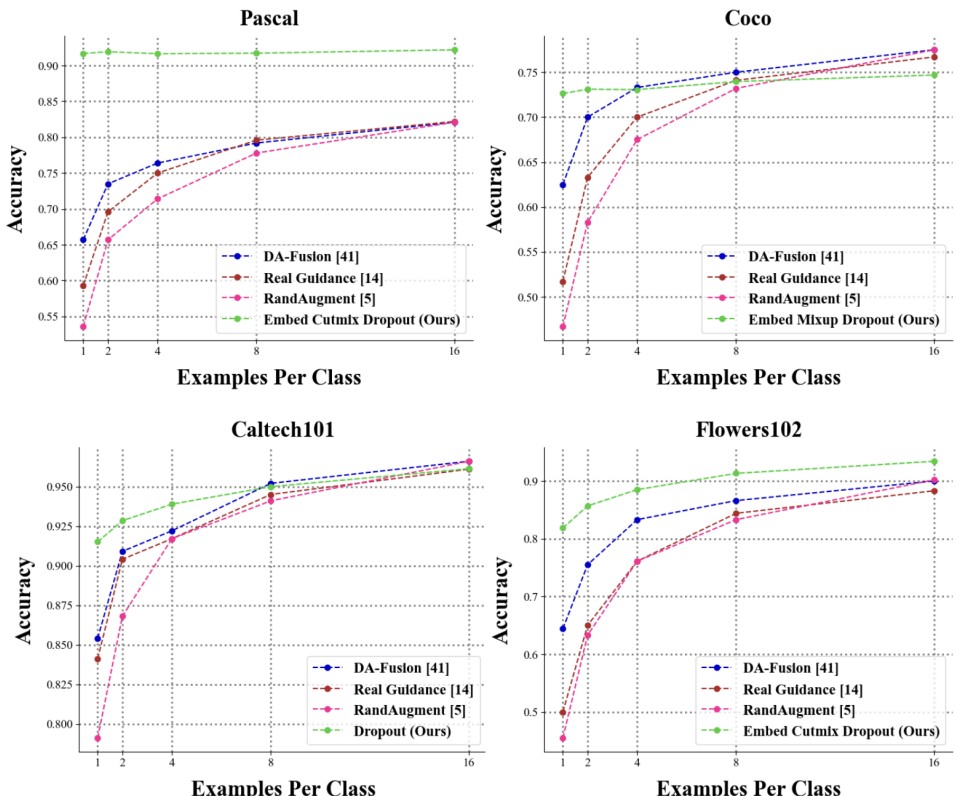

Figure 6: Few-shot classification performance of the best-performing conditioning method compared to existing work on 4 datasets. Augmentation-conditioned generations match or improve accuracy up to +25% over the best-performing existing method, with no training of the diffusion model.

generations achieves up to +10% accuracy across a variety of few-shot classification settings, over diffusion-based data augmentation methods that require fine-tuning of the diffusion model. Utilizing augmentation-conditioned generations as training data also improves over state-of-the-art results on a long-tail, imbalanced classification task.

We find that leveraging existing data augmentations as conditioning information in the diffusion process produces effective synthetic training datasets for various classification tasks, without requiring fine-tuning of the diffusion model. Augmentation-conditioned generations thus enable training image generation at the same cost as general image generation with an off-the-shelf text-to-image model. Conditioning on real training images enables generations to be in-domain with the real image distribution, while the data augmentations introduce visual diversity that enhances the performance of the downstream classifier. We improve classification performance on long-tail and few-shot vision benchmarks by training on our generated images, showing that augmentation-conditioning generates effective training data for a variety of tasks. Augmentation-conditioned generations are a computationally efficient approach to using pretrained diffusion models as training image generators.

**Limitations & Future Work.** Using our conditioned generations as synthetic training data enables strong performance improvements, however there are limitations. The pre-trained diffusion model we use for image generation may include examples from common vision benchmark datasets, such as ImageNet Deng et al. (2009) and COCO Lin et al. (2014), as it is trained on billion-scale Internet data. Previous work has shown that pre-trained diffusion models can memorize training examples, leading to training data leakage Carlini et al. (2023). As future work, we would like to investigate the effect of potential data leakage on the downstream model performance.

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

## A    DROPOUT PROBABILITY'S EFFECT ON IMAGE DIVERSITY

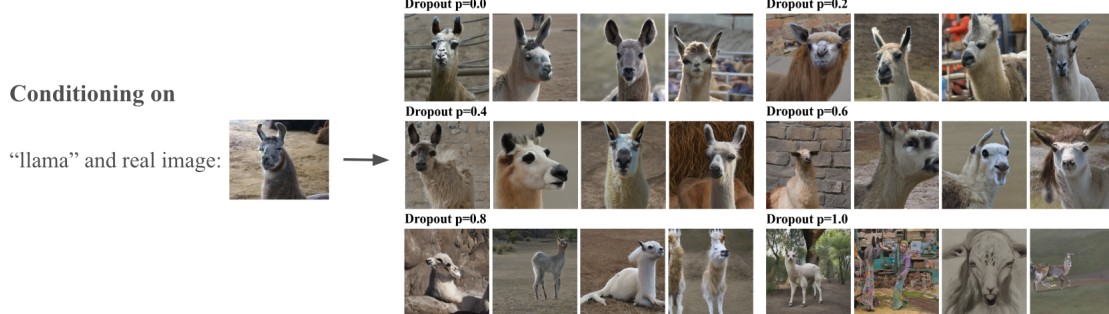

Figure 7: Example generations conditioned on Dropout with various probabilities applied to a real image. $p = 0.0$ is equivalent to conditioning on the original image and generations lack visual diversity. $p = 1.0$ is equivalent to only conditioning on the text class label, resulting in out-of-domain images. Dropout probabilities in the middle yield diverse but in-domain images.

See 7 for Dropout Probability's affect as an image generation hyperparameter. A similar plot of image generations is also shown in Hemmat et al. (2023).

# B    HYPERPARAMETERS AND TRAINING DETAILS

The full set of hyperparameters for image generation and classifier training are given in Table 4.

All experiments were run on A100, A40, and A5500 GPUs on university compute clusters.

| Hyperparameter Name | Value |
| --- | --- |
| **Image Generation** | |
| LDM-v2.1-unCLIP Checkpoint | `stabilityai/stable-diffusion-2-1-unclip` |
| Diffusion Denoising Steps | 30 |
| Diffusion Noise Scheduler | PNDM Scheduler Liu et al. (2022) (default in Hugging-Face) |
| **Section 4.1 Classifier** | |
| Architecture | ResNext50 |
| Learning Rate | 0.2 |
| Momentum | 0.9 |
| Weight Decay | 0.0005 |
| Batch Size | 512 |
| Training Epochs | 150 |
| **Section 4.2 Classifier** | |
| Architecture | ResNet50 |
| Learning Rate | 0.0001 |
| Batch Size | 32 |
| Fine-Tuning Epochs | 50 |

Table 4: Hyperparameters and training configuration details

Results from Sections 4.1.1 and 4.1.2 use the downsized ResNet18 (with the training configuration of Section 4.1) and a 90-class-subset of all 1K ImageNet classes. See code files for names of classes in the 90-class-subset.

# C    INDIVIDUAL FEW-SHOT CLASSIFIER FREE GUIDANCE PLOTS

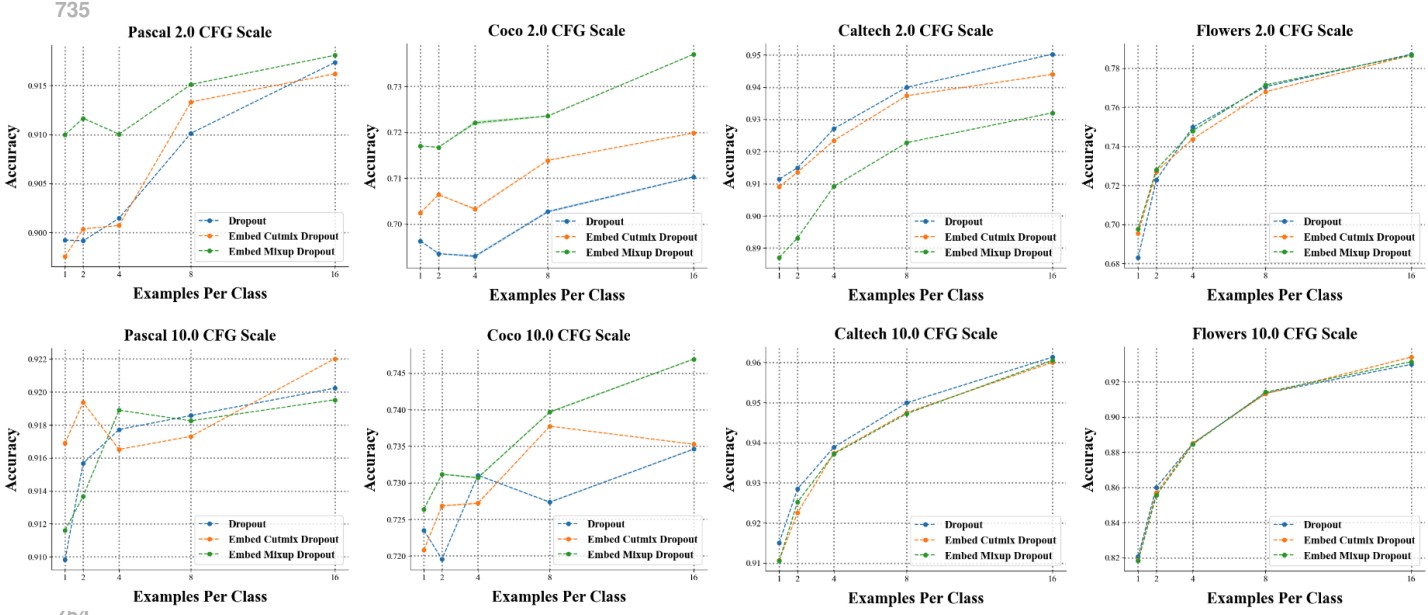

Figure 8: Classifier free guidance scale's affect on few-shot classification performance

