# OpenReview forum: "Augmented Conditioning is Enough for Effective Training Image Generation"
_ICLR.cc/2025/Conference — ICLR 2025 Conference Withdrawn Submission_

### Official Review · Reviewer_nirE · 2024-10-15

**Soundness:** 2
**Presentation:** 3
**Contribution:** 1
**Rating:** 3
**Confidence:** 5

**Summary:**

While synthetic training images generated by diffusion models can be effective for training, diversity and fidelity are challenges. Fidelity can be addressed by image conditioning with few-shot images, but many previous works fine-tune the diffusion model, which can be expensive. The authors propose a frozen alternative to increase the diversity, which conditions the diffusion model not only on few-shot images (done previously) but augmentations (novelty).

They evaluate this across different standard data augmentation techniques, comparing downstream training accuracy and FID to find the most effective augmentation techniques. They also evaluate the effects of CFG scale, compare their method to previous SOTA in long-tailed distributions, and across 4 datasets to previous few-shot SOTA by number of shots.

**Strengths:**

A) Writing is overall quite clear.

B) Experiments are varied across areas (long-tail, few-shot).

C) Method section shows many qualitative examples to supplement the quantitative results in the experimental section.

D) Some of the results show improvement.

**Weaknesses:**

Points are ordered roughly according to my perceived scale, with more important points being listed first.

A) The method itself is quite simplistic from a novelty perspective (simply adding augmentations to the conditioning). I would consider this a strength, if the results were consistent (B) and strong (B, D, E, F) with a clear storyline for effective use-cases (C). However, I do not see this as being the case (see following points for details, as indicated in the corresponding parentheses).

B) The results are mixed. Examples: 1) In table 1, the random image baseline actually has the lowest FID. I do not see this discussed, with lines 323 and 339 pointing out that "the best-performing augmentation-conditioning method has one of the lowest FID scores, supporting our claim...", which is misleading. 2) In table 3, unintuitive bolding hides that your method underperforms in some categories (with LDM t&i, medium is worse and few is tied). 3) In Figure 6, by 16 shots, the novel method is already underperforming on 2 datasets. Once again, the writing does not properly address this.

C) Given the mixed results, there should be an in-depth analysis / explanation to understand when this method is most useful, but this seems to be missing. I want to stress that if done well, this could potentially make up for weakness B.

D) The comparisons with Fill-Up are not clear, as "Ours" does worse but uses less data--it would be better to compare against Fill-Up with the same amount of data as well, otherwise you have not shown that you are beating SOTA.

E) There seems to be some baselines missing in the few-shot section that could strengthen the context. 1) as the ResNet50 is pre-trained, it would be helpful to know the starting accuracy. 2) Figure 6 is missing the random-image baseline included in Table 1.

F) I do not find the CFG scale experiments as adding significant value, although they take up a page in total. They are consistent with previous work, which I don't find unsurprising. While they don't really 'hurt' anything in themselves, they overall weaken the experimental section by taking the space of what could have otherwise been more interesting / surprising / novel results, and in my opinion they water down the impact of the experimental section.

G) In the introduction, it is claimed that methods that fine-tune the diffusion model (Azizi 2023, Trabucco 2023, Shin 2023) are too expensive. However, this is never supported with numbers--it would make the claim stronger to quantify method costs. Especially because these methods have vastly different costs (e.g. in line 134, you claim that Shin 2023 uses textual inversion--this should be less expensive than full fine-tuning, correct? And what about methods that use PEFT?)

H) Table 3 is misleading. The way the sections are split, the entire lower section should be compared. However, Fill-Up is a separate class with more synthetic data, which you are not comparing against. This is not very clear by the visual, and is confusing why the highest numbers are not the ones bolded (as Fill-Up is excluded). Sometimes the bolding seems entirely wrong--e.g. in medium, "ours" is bolded, but LDM (txt and image) is clearly better, and also at comparable synthetic data counts.

**Questions:**

A) In the description of Figure 4, you claim in line 236 that "Augmentation-conditioned generations show more visual diversity in the coloration, pose, and angle of the hamster." Could you please elaborate on what you mean? Because I do not see this as being the case. To me, 1) coloration looks consistent across all images, 2) pose only shows clear differences for CutMix and CutMix-Dropout. All others are primarily face photos staring at the camera and 3) with the same exceptions as 2, angle is primarily straight-on.

B) In table 1, it is shown that the improvements are the most significant on the few classes--this is potentially interesting. Is there some analysis, interpretation, explanation, etc. that could be added on this subject?

---

### Official Review · Reviewer_ySzm · 2024-11-03

**Soundness:** 3
**Presentation:** 3
**Contribution:** 2
**Rating:** 5
**Confidence:** 3

**Summary:**

The author explores a novel method for training image generation, where the proposed approach produces in-domain images with enhanced visual diversity by conditioning the generation process on augmented real training images. This strategy is validated across five established long-tail and few-shot image classification benchmarks, demonstrating consistent improvements over state-of-the-art results.

**Strengths:**

- The problems and the proposed method are clearly presented and easy to understand.
- The proposed method is effective and straightforward in practice, with extensive ablation experiments conducted to support its design.
- The synthesized training data demonstrates strong performance in few-shot classification.

**Weaknesses:**

In Table 3, the Fill-Up method demonstrates higher accuracy than the proposed method. Although the positive correlation between accuracy and training dataset scale is discussed in line 376, it remains unclear whether the proposed method can outperform Fill-Up. Given the computing constraints, I suggest the authors:

- Illustrate the positive relationship between accuracy and dataset scale using data synthesized by the proposed method.
- Provide results for Fill-Up with different, smaller synthetic dataset scales that are more feasible.

**Questions:**

See above weaknesses.

---

### Official Review · Reviewer_mZw1 · 2024-11-04

**Soundness:** 3
**Presentation:** 1
**Contribution:** 2
**Rating:** 3
**Confidence:** 4

**Summary:**

This paper demonstrates that conditioning the generation process on an augmented real image and a text prompt produces effective synthetic datasets. These synthetic datasets benefit downstream tasks, particularly for long-tailed (LT) classification and few-shot classification.

**Strengths:**

- The approach appears valid and effective for LT and few-shot classification, as demonstrated in the experiments. The authors also tested various augmentations.
- Conditioning on both the augmented image and text prompt seems effective for improving performance on classification tasks.
- Experiments on different values of classifier-free guidance (CFG) are interesting, especially regarding how the optimal scale varies by task.

**Weaknesses:**

- The technical novelty of this paper is unclear. The concept of combining both augmented images and text prompts seems useful for LT and few-shot classification but lacks novelty. If this approach is not technically original, the paper should at least show a broad variety of downstream tasks that benefit from it, which it did not.
- The contribution is not clearly articulated. Although it’s evident that the synthetic dataset is effective, it’s unclear for which specific tasks it is most useful. The focus is confined to LT and few-shot classification. Could this approach also aid in other areas, such as image generation? Expanding the application scenarios would improve the paper’s impact.
- LT classification works, particularly those focused on algorithmic improvements, were not compared in the evaluation. While the paper’s approach differs by aiming to improve classification via synthetic data, it’s worth questioning if this is truly beneficial. For example, the paper mentions that fine-tuning for classification is time-consuming. However, both fine-tuning and generating a synthetic dataset have costs - generating 1.16M images seems likely to take more time than fine-tuning.

**Questions:**

- In conclusion, what do you suggest as the best approach among the trials (e.g., Mixup-Dropout, Embed-Mixup-Dropout, Embed-CutMix-Dropout)? In Table 3, the last method performs best, but in Figure 8, Embed-Mixup Dropout seems to lead in many cases. Does the optimal choice depend on the task and dataset? Are there any noticeable patterns?
- Why did you focus specifically on LT and few-shot classification? Couldn’t this approach and synthetic dataset also benefit other discriminative downstream tasks or image generation?
- How long did it take to generate approximately 1 million synthetic images (as in Table 3)? Did you use T=1000 for generation, or T=50? Does the choice of timestep T affect the downstream performance (e.g., classification accuracy)?
- How do you support the claim that your dataset has the highest diversity? In Table 1, the FID scores are provided, but FID does not solely calculate diversity. Also, FID of your approach and the baseline (random images) are nearly the same, with the baseline slightly lower. To substantiate the claim of high diversity, perhaps an additional metric, such as recall, would be helpful.
- In the bottom-right graph of Figure 5, why are the performances of the three methods almost identical? Is there any underlying interpretation for this?

---

### Official Review · Reviewer_CfJe · 2024-11-04

**Soundness:** 2
**Presentation:** 3
**Contribution:** 2
**Rating:** 5
**Confidence:** 4

**Summary:**

This paper focuses on improving the effectiveness of synthetic images generated by text-to-image diffusion models for training image classification models. By introducing “augmentation-conditioning”, the authors leverage real images with data augmentations as conditioning inputs, creating synthetic images that are not only realistic but also visually diverse. This approach enhances downstream classifier performance, particularly in long-tail and few-shot classification settings, without the need for fine-tuning the diffusion model itself. The method was validated across five benchmarks, showing consistent improvements over previous techniques.

**Strengths:**

1. The paper introduces an approach of augmentation-conditioning, which leverages real images with data augmentations to create synthetic images that are both realistic and diverse. This method bridges the domain gap between synthetic and real data, enhancing downstream classification performance without requiring extensive fine-tuning of the diffusion model.
2. The method’s effectiveness is demonstrated across multiple challenging benchmarks, including long-tail and few-shot classification tasks.

**Weaknesses:**

1. The technical novelty of the proposed method seems limited, as it mainly combines existing data augmentations, like Mixup, before inputting images into an existing diffusion model. More discussion of method’s novelty is necessary. Besides, to better demonstrate the effectiveness of the proposed method, it would be beneficial to consider more recent tuning-free approaches for diffusion models, such as [1]. Additional discussion and experiments comparing the superiority of the proposed method with current works would strengthen the paper.

2. While the proposed augmentation-conditioning method focuses only on tuning-free augmentation approaches, there are low-cost tuning methods, such as [2] using LoRA, that also deliver strong performance. Why prioritize tuning-free methods when low-cost tuning options might achieve better results with only a minor increase in computational cost?

3. The application scope of the proposed method appears limited (i.e., primarily for long-tail or few-shot classification), and the experimental validation seems insufficient.

a) In Table 1, the performance gain is more pronounced in the few-shot setting (fewer than 20 images, 55.3 -> 63.5) than in the many-shot setting (100 or more images, 72.4 -> 72.0). This suggests the method may not be as effective in many-shot contexts, even with only 100 images. A clearer explanation of this phenomenon would enhance the method’s credibility.

b) All experiments are conducted on the ImageNet-LT dataset, which alone cannot comprehensively verify the method’s performance. Testing on a wider range of datasets, including general datasets like Tiny-ImageNet-200, CIFAR-100, and full ImageNet as in [1], as well as domain-specific long-tail datasets like CUB-LT and Flower-LT as in [2], would provide more robust evidence.

[1] DIFFUSEMIX: Label-Preserving Data Augmentation with Diffusion Models. CVPR, 2024.

[2] Enhance Image Classification via Inter-Class Image Mixup with Diffusion Model. CVPR, 2024.

**Questions:**

Please refer to Weaknesses.

---

### Note · Authors · 2024-11-23

I have read and agree with the venue's withdrawal policy on behalf of myself and my co-authors.